# PARAMETER-FREE VIDEO SEGMENTATION FOR VISION AND LANGUAGE UNDERSTANDING

## ABSTRACT

The proliferation of creative video content has driven demand for adapting language models to handle video input and enable multimodal understanding. However, end-to-end models struggle to process long videos due to their size and complexity. An effective alternative is to divide them into smaller chunks to be processed separately, and this motivates a method for choosing where the chunk boundaries should be. In this paper, we propose an algorithm for segmenting videos into contiguous chunks, based on the minimum description length principle, coupled with a dynamic programming search. The algorithm is entirely parameter-free, given feature vectors, not requiring a set threshold or the number or size of chunks to be specified. We show empirically that the breakpoints it produces more accurately approximate scene boundaries in long videos, compared with existing methods for scene detection, even when such methods have access to the true number of scenes. We showcase this algorithm in two tasks: long video summarisation, and retrieval-augmented video QA. In both cases, scene breaks produced by our algorithm lead to better performance than existing segmentation methods.

## 1 INTRODUCTION

With the proliferation of streaming services and digital content providers, a large number of movies and television series are being released and made available every year. Automatic approaches to understanding and summarising their content are paramount to enabling users to browse or skim through them, and quickly recall key plot points, characters, and events without the need to rewatch. Aside from practical utility, the complex narrative understanding required in long videos makes them an ideal testbed for the capabilities of large vision language models (LVLMs).

A key step in long video understanding is being able to break the video up into smaller pieces, as this allows LVLMs to process smaller chunks independently, and to selectively focus on the most relevant parts. An earlier line of research Lupatini et al. (1998); Yeung and Yeo (1996); Zabih et al. (1995); Sanchez et al. (1999) focuses on the problem of *scene break detection*, i.e., determining where one scene ends and another begins in a long, narrative video, mostly by placing cuts where pixel differences exceed some threshold. PySceneDetect[1], the widely used Python library follows this idea, converting to HSV channels Ford (1998) and then computing differences between consecutive frames. More recent progress in scene break algorithms has been limited, with only a handful of deep learning models trained on specific domains Liu et al. (2020); Rao et al. (2020); Mun et al. (2022); Ye et al. (2022); Wang et al. (2024c). In this paper, we propose a new scene segmentation algorithm, which we call `MDLSeg`, based on the minimum description length principle. `MDLSeg` does not search for frame differences exceeding some threshold, indeed, it does not require setting a threshold, or the number of scenes, or any parameters at all. Instead, it searches all the different ways of grouping the feature vectors for each frame, and selects the one that can be represented with the fewest number of bits. This encourages having every scene contain frames with feature vectors similar to each other, but also not having too many scenes.

We further demonstrate that scene segmentation is useful for designing modular video understanding systems, i.e., those based on a number of interacting components that separately solve different subtasks. This design differs from recent work (e.g., Song et al. (2024)) aiming to handle longer

---

[1] https://www.scenedetect.com/

video sequences through modifications to transformer memory. Scaling such end-to-end models to full-length movies, remains a significant challenge due to memory constraints and the complexity of extracting useful information from large inputs.[2] We explore the practical utility of `MDLSeg` for two long video understanding tasks: summarisation and question answering. Summarisation has seen significant advances thanks to large models with extended context windows and the design of methods which rely on dividing the input into chunks Chen et al. (2023a); Pang et al. (2023); Chang et al. (2023); Papalampidi and Lapata (2023); Mahon and Lapata (2024). We show that `MDLSeg` can boost the performance of such methods. We also apply `MDLSeg` to long video question answering Ataallah et al. (2024), incorporating it into a retrieval-augmented generation (RAG) QA pipeline. Given a long video and a question about its content, we use `MDLSeg` to segment it into scenes and retrieve the most relevant one based on the query; we generate a textual description of the retrieved scene and use it to answer the question. In both cases, experimental results demonstrate that scene breaks produced by our algorithm lead to better downstream performance than existing methods for video segmentation. Our contributions can be summarised as follows:

- A novel scene-break detection method, which is parameter-free given the frame features.
- Empirical results showing that scene breaks from our method are more accurate than existing methods or baselines, even when the latter have access to the *true* number of scenes.
- A demonstration of how our method can improve the downstream performance of modular systems, as part of hierarchical movie summarisation and retrieval-augmented VQA.

## 2 RELATED WORK

**Video Segmentation**    A simple method for scene break detection is to follow differences between consecutive frames. The popular **PySceneDetect** library computes a histogram of pixel intensities for each frame in HSV space Ford (1998), and then computes the absolute difference between the histograms for consecutive frames, and places a scene break where this difference exceeds some user-set threshold. Some work imposes a temporal constraint on k-means Berhe (2021) or incorporates temporal distance information into a hierarchical clustering algorithm Yeung and Yeo (1996). Rotman et al. (2017a) also propose a dynamic programming search for scene partitioning, but optimise a different objective from ours that does not employ MDL. A number of deep-learning-based methods have also been proposed. **Lgss** Rao et al. (2020) detects scene boundaries by modeling local-global shot similarities using a Siamese network. **SCRL** Wu et al. (2022) learns scene-consistent shot representations via self-supervised contrastive learning. **HEM** Cheng et al. (2024) segments long videos into events and models them hierarchically for memory-augmented LLM-based understanding. **bassl** Mun et al. (2022) introduces boundary-aware self-supervised pretext tasks to improve scene segmentation. **neighbornet** Tan et al. (2024) redefines shot similarities using semantic and temporal neighbors via graph convolution. **scene-tile** Wang et al. (2024c) segments videos by detecting low-similarity valleys in adjacent frame embeddings from a ViT. `MDLSeg` differs from all of these in not requiring any training and in being based on MDL.

**Video Understanding**    The problem of generating descriptions for videos has received significant attention in the literature. Traditional approaches often extract features from individual frames and fuse them into a single feature vector to generate a textual description (Zhang et al., 2021; Pan et al., 2020; Ye et al., 2022). SwinBERT (Lin et al., 2022) introduces an end-to-end video network that samples frames densely, avoiding the need for image-based encoders. Similarly, Lei et al. (2020) generate descriptions for short videos with a memory-augmented transformer. Some work aims to summarise short videos, a task referred to as video captioning. Example methods include using a two-stream CNN Sridevi and Kharde (2020), developing a bidirectional model that uses both video and audio to produce video captions Seo et al. (2022), and proposing a single masked transformer objective to detect and then caption all events Zhou et al. (2018b).Unsupervised pretaining has also been explored, e.g., by Yang et al. (2023), who train a video-captioning model using transcribed utterances as pseudo-captions. Systems based on large proprietary models have also been proposed for longer videos Zhang et al. (2024); Lin et al. (2023) with multiple modules, including visual GPT-4 and PysceneDetect for scene breaks. Wu et al. (2024) prompt an LLM to predict scene breaks from *transcribed* speech and captions, which are then used for video question-answering.

---

[2]At $1,024 * *2$ frame size, and 10fps, a 75min movie would consume over 500GB as a 4d 32-bit float tensor.

**Long-form Summarisation and QA**   Much of the work just described is suitable only for short videos (Chen and Dolan, 2011; Xu et al., 2016), ranging from ∼10s in length to 5 minutes (Zhou et al., 2018a) at the upper end. Recent work has started to leverage segmentation to address the task of understanding much longer videos. Chen et al. (2023b) propose Movies2Scenes, a method that uses movie metadata to learn video representations for long movies divided into scenes, though it relies on predefined scenes based on shot transitions rather than semantically meaningful boundaries. Papalampidi et al. (2021) summarise full-length movies by creating shorter videos containing their most informative scenes which they assume to be 'turning points' (i.e., key events in a movie). Other work produces text summaries of TV shows, by converting visual features into embeddings alongside word embeddings from the transcript Papalampidi et al. (2021) or by converting the video to text, and then treating it as a text-only problem Mahon and Lapata (2024). In a similar vein, long-form question answering explores the ability of models to understand videos longer than five minutes. Existing approaches improve model capacity to handle longer context windows through architectural modifications Song et al. (2024) or by designing modular systems which either translate the video into text and then extract important information from it Wu et al. (2024) or segment the input and rely on retrieval to isolate important segments for question answering Ataallah et al. (2024).

Our scene segmentation algorithm, `MDLSeg`, is data- and parameter- free, given the frame-features. It works with any type of long-form video, it does not depend on written transcripts or screenplays, which are not always available (e.g., video providers do not have access to screenplays unless they have produced the content themselves), and does not require setting the number of scenes.

## 3   `MDLSeg`: Minimum Description Length-based Segmentation

Scene segmentation is essentially a clustering problem with the additional constraint that each cluster must be contiguous. Intuitively, a good cluster must fulfill two objectives: each point should be close to its cluster centroid, and there should not be too many clusters. Normally, these two objectives are not quantified in the same way, so it is difficult to trade off one against the other. However, MDL allows us to quantify both in the same units–bits–so that they can be directly compared, and their sum can be minimized. In general, this optimisation problem does not have a straightforward solution, but part of our unique contribution is that, when coupled with the contiguity constraint, minimising the description length in fact admits an efficient exact, or near-exact, solution.

`MDLSeg` computes a partition of the visual features from each keyframe, with the constraint that each subset in the partition must be contiguous. There are two parts to the algorithm: the *definition of a cost* for a particular partition into scenes, and the *search* for the partition that minimizes this cost. The first part, the cost definition, is formulated using the minimum description length principle, which claims the correct representation of the data is the one using the fewest bits Grünwald (2007). We assume that the vectors for each scene are encoded with respect to their collective mean. That is, for each scene in the given partition, we calculate the mean and covariance matrices of all vectors in that scene, and hence, the probability of each vector, $p(v)$, under the multivariate normal distribution with these parameters. The Kraft-McMillan inequality (Kraft, 1949; McMillan, 1956) then determines that under the optimal encoding, the number of bits needed to represent $v$ is $-\log_2 p(v)$. The sum of this value across all $N$ vectors $v$ in the video, plus the number of bits to represent the means and covariances themselves, gives the total bitcost for a given partition. Both the mean and the covariance require $dm$ bits (we use diagonal covariances), where $d$ is the dimensionality, and $m$ is the floating point precision. We choose the precision based on the data as the smallest value that allows it to be represented exactly. Partitions with more scenes require more bits for the mean vectors, but also have mean vectors that better cover the keyframe features, leading to decreased $-\log_2 p(v)$ on average. This trade-off encourages a partition with neither too few nor too many scene breaks.

The second part, the search for the minimizer of the above cost, can be solved exactly using dynamic programming. Let $B(i,j)$ be the cost of having a single scene that runs from keyframes $i$ to $j$, and let $C(i,j)$ be the minimum cost of all keyframes from $i$ to $j$, under all possible partitions. Then, we have the following recurrence relation, which allows iteratively computing and caching $C(i,N)$ for $i = N-1, \ldots, 0$:

$$C(i,j) = \min_{i+1 \leq k \leq j} B(i,k) + C(k,j) \,. \tag{1}$$

The full procedure for `MDLSeg` is shown in Algorithm 1 and an example is depicted in Figure 1.

**Algorithm 1** Video Scene Partitioning

**Require:** Video file
 1: Extract keyframes, $kf_0, \ldots, kf_N$
 2: Extract visual features $v_0, \ldots, v_N$ from each keyframe
 3: $L \leftarrow$ maximum scene length
 4: $B \leftarrow N \times N$ empty matrix                    $\triangleright$ $B[i, j]$ will hold the cost of a scene from $v_i$ to $v_j$
 5: $d \leftarrow$ dimensionality of $v_i$
 6: $m \leftarrow$ floating point precision of $v_i$
 7: **Cost Definition:** Compute and store costs for all possible scenes
 8: **for** $i = 0$ **to** $N - L$ **do**
 9:     **for** $j = i$ **to** $i + L$ **do**
10:         $\mu \leftarrow \frac{1}{j-i} \sum_{k=i}^{j} v_k$
11:         $\Sigma \leftarrow$ empirical covariance matrix of $v_i, \ldots, v_j$
12:         $C \leftarrow 2dm$                           $\triangleright$ bitcost of the parameters themselves
13:         **for** $k = i$ **to** $j$ **do**
14:             $p(v_k) \leftarrow \frac{1}{(2\pi)^{d/2}|\Sigma|^{1/2}} \exp\left(-\frac{1}{2}(v_k - \mu)^\top \Sigma^{-1}(v_k - \mu)\right)$
15:             $C \leftarrow C - \log p(v_k)$
16:         $B[i, j] \leftarrow C$
17: **Search:** Minimize the bitcost by dynamic programming
18: $C \leftarrow B$                                    $\triangleright$ will hold optimal costs
19: $P \leftarrow N \times N$ matrix of empty sets                    $\triangleright$ will hold optimal partitions
20: **for** $i = N - 1$ **to** $0$ **do**
21:     **for** $j = i$ **to** $\min(N, i + L)$ **do**
22:         **if** $B[i, j] + C[j, N] < C[i, N]$ **then**
23:             $C[i, N] \leftarrow B[i, j] + C[j, N]$
24:             $P[i, N] \leftarrow P[i, j] \cup \{j\} \cup P[j, N]$
25: **return** Optimal scene partition, $P[0, N]$

**Computational Complexity.** For each $i = N - 1, \ldots, 0$, `MDLSeg` computes the best point for the first split based on previously cached costs of smaller segments. This runs in $O(N^2)$ because the number of partition points to check equals $N - i$ for each segment for each $1 \leq i \leq N$. By imposing a fixed threshold of the maximum number $L$ of keyframes in a scene, this becomes $O(N)$. In our experiments, we find that setting $L$ so that the maximum scene length is about 10 minutes does not affect the solution. i.e., produces the same segmentation for all videos in our datasets as leaving $L$ unset. This maximum scene length is not a parameter of the algorithm itself, but merely one that allows it to run more quickly if a value is known. If a user does not set this parameter, the algorithm is still relatively fast, and, either way, almost all the runtime is for extracting the visual feature vectors. The algorithm itself takes a couple of seconds for a full movie when $L$ is set. Empirically, the speed is similar to PySceneDetect when $L$ is set (see Section 6), and up to 20% or 30s longer when not set.

## 4 DOWNSTREAM TASKS

To showcase the utility of `MDLSeg` for long video processing, we engineer two modular systems for video summarisation and question answering which we describe below.

**Long Video Summarisation** We apply a speaker diarization model to obtain a transcript with numeric speaker IDs, which we augment with video captions (Peng et al., 2023) from three evenly spaced keyframes from that screen. By matching the utterance times with the keyframe timestamps, we insert the `MDLSeg` scene breaks into the transcript. To assign character names to transcribed dialogue, for each movie, we first create a database of actors faces and their character names (scaped from the movie's IMDB page). For each scene, and for each character in our name bank, we define the cost of putting that character name in that scene as the minimum distance between an image of that character's face, and a face from any keyframe from the scene (the whole process takes <1s for a full movie). The cost of assigning a character to a speaker ID, is then the sum of such costs across all scenes containing that speaker ID. Assigning speaker IDs to names then reduces to the

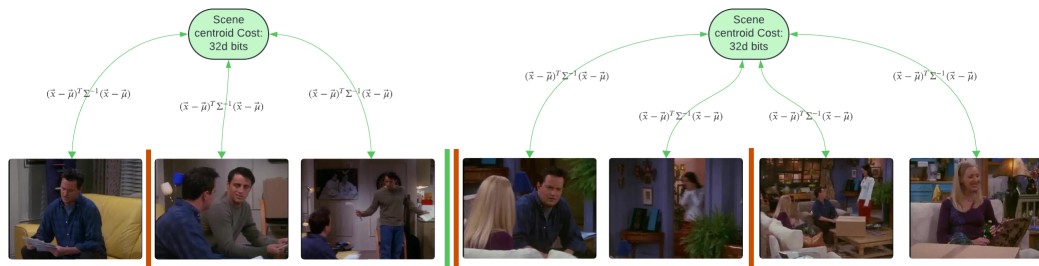

Figure 1: The `MDLSeg` cost of a given partition is the sum of the bitcost for all centroids, and the encoding cost of each frame with respect to its assigned centroid. The former gives $32d$ bits, under 32-bit precision, for each green ellipse, where $d$ is the dimension of the feature space. The latter is the squared Mahalanobis length of each arrow, i.e., between each feature vector $x$, and its centroid $\mu$. The overall partition is chosen to have neither too few scenes, which would give very long arrows, nor too many, which would give many centroid costs. PySceneDetect and other frame-difference methods place breaks (shown in red) where the difference between consecutive frames exceeds a threshold. This misses global structure and places spurious breaks where shots change briefly within scenes.

linear sum assignment problem, which can be solved efficiently using the Kuhn-Munkres algorithm (Kuhn, 1956; Munkres, 1957). We provide additional details in Appendix E.

We adopt a hierarchical summarisation approach (Pang et al., 2023; Chang et al., 2023), as it has proven particularly effective for handling long inputs that are challenging for end-to-end systems. We thus first summarise the transcript dialogue of each scene. We next take the resulting sequence of scene summaries, and their corresponding visual descriptions that were added to the pseudo-transcript by the image captioning model, and summarise them with a text-only model to produce a final summary for the entire movie (see Appendix Figures 2a and 3 for details). The summarisation model is implemented using a widely-used open-source LLM library (Dubey et al., 2024) with zero-shot prompting. We believe our modular system to be novel in its own right, as it requires only video input, figuring out by itself who is speaking and what they are doing, which is taken for granted in previous work Papalampidi et al. (2024); Mahon and Lapata (2024) focusing on screenplays and manual transcripts. However, it is not the focus of the present paper.

**Retrieval-Augmented Video Question Answering** We further apply our scene break algorithm in the task of retrieval-augmented video question-answering. First, we segment the input video with `MDLSeg`. Then, using the timestamps in the transcript, we gather the corresponding text for each scene. Next, we use a vision-to-text model to produce textual descriptions of the video from each scene, so that each scene then consists of a segment of the transcript and a text description of the video. Then we compute feature vectors for each scene, which we use for retrieval. Given a question, we retrieve the scene with the greatest cosine similarity to the question (questions and scenes are represented in the same multimodal feature space) and present it as input to a text-only model, which then produces an answer based on the text from the scene and the question. (A sketch of this modular approach to TVQA RAG is illustrated in Appendix B Figure 2b.) We do not claim novelty in this modular approach, but rather use it to assess the usefulness of the scene breaks from `MDLSeg`.

## 5 EXPERIMENTAL SETTING

**Datasets** For evaluating the scene accuracy directly, we use two datasets designed for this purpose, with human annotated scene breaks: Open Video Scene Detection (OVSD) Rotman et al. (2017b), which consists of 16 free online films of varying styles and genres, including animated, action, drama and children's, ranging from 10 to 90 minutes, and BBC Earth Baraldi et al. (2015), consisting of 11 50 minute episodes of the Planet Earth documentaries. The latter has five sets of annotations, occasionally showing substantial disagreement, so for each method, we report both the mean score compared to all annotators, and the max score, compared to the closest matching annotator.

For the summarisation task, we use the recently released MovieSum dataset (Saxena and Keller, 2024), consisting of screenplays and gold summaries. We were able to obtain corresponding videos

for 175/200 movies in the test set. The remaining 25, we discarded. These movies span multiple fiction genres: drama, action, thriller, comedy, horror, etc. They have an average run time of 118min (range 84–228), with release dates ranging from 1950 to 2023. Gold summaries average 635 words in length. The mean number of scenes in the gold script is 151. We follow recent work Ataallah et al. (2024) in evaluating long video understanding on the TVQA dataset Lei et al. (2018) which consists of videos and accompanying timestamped transcripts from 924 episodes from six TV shows. These shows span 3 genres: 1) sitcoms: The Big Bang Theory, How I Met Your Mother, Friends, 2) medical dramas: Grey's Anatomy, House, 3) crime drama: Castle. Further stats on the dataset are in Appendix I. Neither task requires training inputs, because all stages of our pipeline are zero-shot.

**Implementation Details** Keyframes are extracted as FFMPEG I-frames. The full command is given in Appendix F. As discussed in Section 3, we cap the number of keyframes in a scene to $L = 300$, which roughly corresponds to a 10 minute maximum scene length. Visual features are extracted using CLIP (Radford et al., 2021), specifically 'CLIP-ViT-g-14-laion2B-s12B-b42K' on an 8GB Nvidia RTX 200. In Appendix A, we show results for other feature extractors, which are broadly similar. For the downstream tasks, the speaker diarization model is WhisperX (Bain et al., 2023), an extension of Whisper which can perform speaker diarization and accurate utterance timestamping. The text model for both tasks is Llama 3.1 70B (Touvron et al., 2023). The vision-to-text model is Llava-NeXT Li et al. (2024b), which fine-tunes Llava Liu et al. (2024), on interleaved image-text data and multimodal instruction tuning. We instruct summaries to be a maximum of 635 words (the test set mean), and truncate to 650 words. The feature vectors for VQA retrieval are from InternVideo Wang et al. (2024b), a video foundation model trained using masked video modeling, crossmodal contrastive learning, and next token prediction. We select the multiple-choice answer indicator with the greatest logit value as the answer. For PySceneDetect, we use the default threshold of 27.

**Evaluation Metrics** To directly measure the accuracy of our scene detection method, we use three metrics commonly used in topic segmentation: $P_k$ Beeferman et al. (1997), WindowDiff Pevzner and Hearst (2002), and differential edit distance (ded; Sidiropoulos et al. (2012)), as well as standard partition quality metrics: cluster accuracy (acc), adjusted Rand index (ari), and normalized mutual information (nmi), as defined in Mahon and Lukasiewicz (2024). For summarisation human evaluation is extremely labor-intensive, costly, and difficult to design Krishna et al. (2023). As there is no single agreed-upon metric, we report several complementary metrics aimed at assessing different aspects of summary quality. **Rouge** (Lin, 2004) assesses informativeness against the gold summaries (we report Rouge-2 and RougeL-Sum); **PRISMA** (Mahon and Lapata, 2024) measures factual precision and recall with respect to the gold summary; we use GPT4-turbo for both fact extraction and evaluation stages; **SummaC** (Laban et al., 2022) uses NLI to measure consistency between the input document (gold screenplay) and generated summary; we use the SummaCConv version with 50 evenly-spaced bins; **AlignScore** (Zha et al., 2023) scores the 'informational alignment' between the source (gold screenplay) and the generated summary; we use the base-model checkpoint provided by the authors, and the recommended 'nli' setting with sentence chunk splitting. For the VQA task, we simply report accuracy as the questions are all multiple choice.

## 6 RESULTS

**Scene Detection** Table 1 compares the accuracy of the partitions from `MDLSeg` against eight comparison models. **Uniform**, divides into equal length scenes of number equal to the mean number on each dataset. **Uniform oracle** divides uniformly into the true number of scenes; **kmeans** and **Gaussian Mixture Model** (GMM) cluster the feature vectors and place a scene break between neighbouring time points with different cluster labels. The others, **psd** (PySceneDetect) **Yeung96** Yeung and Yeo (1996), **berhe21** Berhe (2021), **Lgss** Rao et al. (2020), **SCRL**, Wu et al. (2022), **HEM** Cheng et al. (2024), **bassl** Mun et al. (2022), **neighbornet** Tan et al. (2024), and **scene-tile** Wang et al. (2024c) are all existing scene segmentation methods (described in Section 2)

`MDLSeg` produces the most accurate segmentations on all datasets and metrics, surpassing both the other algorithm-based methods and the deep learning-based methods, even including neighbornet, which is trained using labelled scene breaks using the same dataset. The runtime of `MDLSeg` is as fast or faster than existing algorithmic methods. Among the deep learning-based methods, those that essentially just perform a forward pass, such as SCRL and HEM, have the fastest inference time of

Table 1: Scene break accuracy on datasets with manually annotated breaks. On the BBC dataset, we report scores against the best matching annotator of the five annotations per episode (bbc-max), and the mean score across all five annotators (bbc-mean). Best results are **in bold**.

| | | acc ↑ | nmi ↑ | ari ↑ | $P_k$ ↑ | winddiff ↓ | ded ↓ | runtime | training | params. |
|---|---|---|---|---|---|---|---|---|---|---|
| OVSD | unif | 53.18 | 69.93 | 33.12 | 66.30 | 59.87 | 57.95 | 0.00 | No | $k$ |
| | unif-oracle | 54.00 | 72.25 | 38.18 | 72.28 | 50.20 | 52.54 | 0.00 | No | $k$ |
| | kmeans | 49.22 | 62.09 | 11.12 | 38.28 | 467.04 | 76.74 | 127.00 | No | $k$ |
| | GMM | 49.73 | 62.37 | 12.10 | 39.00 | 455.87 | 76.00 | 127.04 | No | $k$ |
| | berhe21 | 58.11 | 71.58 | 30.52 | 65.18 | 80.89 | 56.67 | 127.08 | No | $k$ |
| | psd | 47.35 | 62.37 | 35.42 | 60.56 | 155.77 | 66.28 | 130.31 | No | thresh. |
| | yeung96 | 2.87 | 40.61 | 0.46 | 5.21 | 1080.11 | 98.53 | 143.87 | No | thresh. |
| | scene-tile | 48.30 | 64.65 | 23.48 | 54.31 | 225.03 | 62.51 | 121.34 | No | thresh. |
| | lgss | 52.61 | 36.77 | 12.25 | 79.98 | 57.54 | 63.18 | 4.62 | Yes | None |
| | scrl | 30.83 | 47.95 | 11.81 | 38.37 | 325.44 | 79.29 | 3.11 | Yes | None |
| | HEM | 61.20 | 62.72 | 38.41 | 78.14 | 50.49 | 49.44 | 14.21 | Yes | $k$ |
| | bassl | 52.69 | 44.02 | 23.97 | 76.72 | 69.32 | 56.19 | 126.52 | Yes | None |
| | neighbornet | 59.40 | 62.73 | 34.92 | 76.25 | 46.90 | 52.01 | 21.02 | Yes | None |
| | MDLSeg | **63.37** | **72.58** | **45.13** | **78.39** | **42.58** | **42.99** | 127.86 | No | None |
| BBC-max | unif | 54.62 | 81.53 | 41.34 | 67.65 | 51.09 | 49.95 | 0.00 | No | $k$ |
| | unif-oracle | 54.07 | 81.46 | 40.96 | 73.12 | 44.21 | 49.84 | 0.00 | No | $k$ |
| | kmeans | 54.35 | 77.26 | 20.75 | 46.80 | 242.88 | 65.59 | 20.62 | No | $k$ |
| | GMM | 62.42 | 83.12 | 43.94 | 66.41 | 54.76 | 45.96 | 20.65 | No | $k$ |
| | berhe21 | 53.69 | 76.94 | 20.10 | 46.00 | 245.51 | 66.23 | 20.68 | No | $k$ |
| | psd | 53.23 | 77.93 | 33.97 | 67.75 | 67.53 | 60.66 | 86.45 | No | thresh. |
| | yeung96 | 18.60 | 71.24 | 2.36 | 29.07 | 525.48 | 88.86 | 102.83 | No | thresh. |
| | scene-tile | 52.52 | 77.96 | 32.94 | 56.74 | 123.05 | 56.21 | 19.18 | No | thresh. |
| | lgss | 48.62 | 52.18 | 20.08 | 81.97 | 37.98 | 64.99 | 2.01 | Yes | None |
| | scrl | 60.43 | 82.50 | 44.72 | 58.68 | 85.24 | 46.73 | 1.15 | Yes | $k$ |
| | HEM | 61.53 | 80.05 | 52.82 | 73.55 | 56.67 | 42.64 | 2.12 | Yes | $k$ |
| | bassl | 54.49 | 78.29 | 42.02 | 67.89 | 79.56 | 51.09 | 516.46 | Yes | None |
| | neighbornet | 54.59 | 80.93 | 41.97 | 70.17 | 49.26 | 49.78 | 50.03 | Yes | None |
| | MDLSeg | **69.49** | **85.80** | **60.75** | **83.42** | **26.54** | **35.78** | 21.53 | No | None |
| BBC-mean | unif | 50.59 | 79.16 | 35.91 | 64.43 | 61.83 | 54.70 | 0.00 | No | $k$ |
| | unif-oracle | 48.82 | 79.37 | 35.76 | 64.84 | 60.65 | 54.46 | 0.00 | No | $k$ |
| | kmeans | 51.88 | 73.52 | 16.80 | 42.13 | 257.29 | 69.83 | 20.62 | No | $k$ |
| | GMM | 58.08 | 80.39 | 37.76 | 64.66 | 60.78 | 51.60 | 20.65 | No | $k$ |
| | berhe21 | 51.12 | 73.25 | 16.17 | 41.36 | 260.02 | 70.67 | 20.68 | No | $k$ |
| | psd | 47.01 | 72.67 | 26.69 | 66.07 | 70.43 | 66.96 | 86.45 | No | thresh. |
| | yeung96 | 14.63 | 67.22 | 1.46 | 23.21 | 542.21 | 91.55 | 102.83 | No | thresh. |
| | scene-tile | 49.70 | 75.43 | 28.39 | 55.16 | 126.94 | 59.26 | 19.18 | No | thresh. |
| | lgss | 44.97 | 49.25 | 15.29 | 74.95 | 56.04 | 70.42 | 2.01 | Yes | None |
| | scrl | 56.75 | 80.54 | 40.40 | 58.68 | 91.41 | 50.91 | 1.15 | Yes | None |
| | HEM | 56.65 | 78.65 | 44.79 | 68.24 | 66.62 | 48.97 | 2.12 | Yes | $k$ |
| | bassl | 50.17 | 76.72 | 36.38 | 63.98 | 87.35 | 55.25 | 516.46 | Yes | None |
| | neighbornet | 50.56 | 79.04 | 37.64 | 66.83 | 59.74 | 54.19 | 50.03 | Yes | None |
| | MDLSeg | **66.13** | **83.66** | **54.96** | **77.86** | **42.40** | **40.06** | 21.53 | No | None |

all methods tested, however, this does not take into account the time for pretraining which very likely outweighs any savings in inference time. Those with more complicated inference procedures, (bassl and neighbornet), are a similar speed to MDLSeg (and also require lengthy train time).

Our summaries obtain the highest scores, across all metrics. The improvement is largest for the fact-based metrics of PRISMA (comprised of fact-prec and fact-rec), and AlignScore. Otter and mahon24 especially struggle with such metrics. We find that Otter is mostly able to capture surface-level detail, with descriptions such as "a woman gets out of a car and goes into a building", but is unable to construct a narrative such as "a woman drives to the bank to deposit the money", so ends up capturing very little of the plot. The low scores of 'mahon24', on the other hand, are largely due to the older, smaller backbone model (BART; Lewis et al. 2020), which often becomes decoupled from the input and produces unrelated output, highlighting the importance of incorporating current LLMs into video summarisation models.

| | acc ↑ | nmi ↑ | ari ↑ | $P_k$↑ | winddiff ↓ | ded ↓ | runtime | training | params. |
|---|---|---|---|---|---|---|---|---|---|
| unif | 53.18 | 69.93 | 33.12 | 66.30 | 59.87 | 57.95 | 0.00 | No | $k$ |
| unif-oracle | 54.00 | 72.25 | 38.18 | 72.28 | 50.20 | 52.54 | 0.00 | No | $k$ |
| kmeans | 49.22 | 62.09 | 11.12 | 38.28 | 467.04 | 76.74 | 127.00 | No | $k$ |
| GMM | 49.73 | 62.37 | 12.10 | 39.00 | 455.87 | 76.00 | 127.04 | No | $k$ |
| berhe21 | 58.11 | 71.58 | 30.52 | 65.18 | 80.89 | 56.67 | 127.08 | No | $k$ |
| psd | 47.35 | 62.37 | 35.42 | 60.56 | 155.77 | 66.28 | 130.31 | No | thresh. |
| yeung96 | 2.87 | 40.61 | 0.46 | 5.21 | 1080.11 | 98.53 | 143.87 | No | thresh. |
| scene-tile | 48.30 | 64.65 | 23.48 | 54.31 | 225.03 | 62.51 | 121.34 | No | thresh. |
| HEM | 61.20 | 62.72 | 38.41 | 78.14 | 50.49 | 49.44 | 14.21 | Yes | $k$ |
| lgss | 52.61 | 36.77 | 12.25 | 79.98 | 57.54 | 63.18 | 4.62 | Yes | None |
| scrl | 30.83 | 47.95 | 11.81 | 38.37 | 325.44 | 79.29 | 3.11 | Yes | None |
| bassl | 52.69 | 44.02 | 23.97 | 76.72 | 69.32 | 56.19 | 126.52 | Yes | None |
| neighbornet | 59.40 | 62.73 | 34.92 | 76.25 | 46.90 | 52.01 | 21.02 | Yes | None |
| MDLSeg | **63.37** | **72.58** | **45.13** | **78.39** | **42.58** | **42.99** | 127.86 | No | None |

**Summarisation** In Table 2, we evaluate the summaries generated by a hierarchical method using `MDLSeg` scene breaks as input (see Figure 2a). We benchmark against three baselines using Llama 3.1 70B as their backbone: **name-only** uses the parametric knowledge of the LLM without any content input, e.g., the prompt is 'Summarize *The Silence of the Lambs*';[3] **full script** uses the entire gold screenplay as input, and for **whisperX** the input is the WhisperX transcript. We also compare to two existing models: **Otter**(Li et al., 2023), an end-to-end video description model based on video-llama2; and the modular model of Mahon and Lapata (2024) which takes videos and gold screenplays as input. For Otter, we divide the input video into 3min chunks, and combine the model description of each chunk.

Prompting with only the movie name performs reasonably well, confirming that Llama3.1 has significant parametrically

Table 2: Summarisation results on MovieSum. Top 3: baselines we implement. Middle 2: existing long-form multimodal summarisation methods. Bottom 4: ablation studies: 'w/o names' does *not* replace speaker IDs with character names using our assignment method; 'w/o scene breaks' summarises the screenplay in one pass without scenes breaks; 'unif-breaks' breaks uniformly instead of using `MDLSeg`. f-prec, f-rec, and align abbreviate fact-precision/recall and AlignScore. Best results are **in bold**.

| | r2 | rl-sum | f-prec | f-rec | PRISMA | align | summac |
|---|---|---|---|---|---|---|---|
| name-only | 9.53 | 41.17 | 50.40 | 43.04 | 44.16 | 53.11 | 26.57 |
| full script | 9.32 | 39.94 | 48.77 | 52.73 | 49.05 | 68.59 | 25.83 |
| whisperX | 9.22 | 39.94 | 46.73 | 53.65 | 48.00 | 68.57 | 25.86 |
| Otter | 3.06 | 26.73 | 11.67 | 8.95 | 5.18 | 45.90 | 24.37 |
| mahon24 | 2.79 | 19.97 | 23.16 | 23.19 | 19.28 | 46.32 | 26.97 |
| w/o breaks | 8.45 | 36.82 | 48.32 | 51.79 | 49.99 | 71.95 | 26.31 |
| unif-breaks | 8.45 | 36.82 | 46.58 | 50.69 | 48.11 | 57.62 | 25.73 |
| psd-breaks | 2.15 | 15.18 | 15.93 | 27.38 | 16.12 | 52.29 | 32.82 |
| ours | 10.32 | **44.50** | **55.24** | **54.77** | **53.57** | **72.76** | **27.24** |

stored information about these movies. However, these summaries are short, and when asked for a longer summary, the model repeats the same information over and over. Surprisingly, giving the full gold screenplay as input does not produce better summaries than our method or than some other baselines. This shows there is still difficulty in summarising very long text inputs. When prompted with the name only, Llama-3.1 very likely effectively regurgitates an existing online summary. However, when the prompt also includes the transcript or screenplay itself, Llama tries to actually summarise the information given, during which it can make mistakes. In Appendix H we provide example summary output for the modular method using `MDLSeg` and the best-performing comparison methods.

eTable 2 (third section) also shows ablations on different components (see Figure 2a). **'w/o names'** omits replacing speaker IDs with character names. This causes summary quality to drop, showing the usefulness of name assignment to downstream summaries. **'w/o scene breaks'** feeds the entire pseudo-screenplay to Llama 3.1, instead of using `MDLSeg` to split into scenes and summarising hierarchically. The drop in summary performance in this setting shows the effectiveness of the

---

[3]Precise prompts are given in Appendix G.

Table 3: Accuracy on the TVQA-long dataset. Left: segmentation+retrieval methods–`MDLSeg` (ours), PySceneDetect (psd), unif, GMM–plus Llama3.1-70b. Goldfish and Llama-vid are two existing long TVQA models with their own chunking method. Splitting the scenes using `MDLSeg` gives higher QA accuracy than splitting uniformly or with GMM or PSD. Right: end-to-end LVLMs. The bottom row shows results without the video input. The high scores of the LVLMs suggest contamination.

|  | ours | psd | GMM | unif | no-splits | goldfish | llama-vid | qwen-vl | lava-onevision | pllava | tarsier |
|---|---|---|---|---|---|---|---|---|---|---|---|
| w/ input | 40.92 | 33.68 | 20.05 | 39.99 | 20.09 | 41.78 | 26.86 | 38.70 | 39.53 | 34.84 | 34.09 |
| w/o input | 19.51 | 19.51 | 19.51 | 19.51 | 19.51 | 19.51 | 19.51 | 36.85 | 34.99 | 22.19 | 22.47 |

hierarchical summarisation method enabled by the scene breaks obtained from `MDLSeg`. **'unif-breaks'** and **'psd-breaks'** still adopt the hierarchical summarisation method, but instead of using `MDLSeg`, split scenes into uniform chunks of 250 tokens (the mean from `MDLSeg`) or split into the scenes from PySceneDetect. These settings also degrade summary quality, showing that the higher accuracy from our segmentation method (Table 1), also leads to improved downstream summaries.

**Retrieval-augmented Video Question Answering**   Table 3 shows model accuracy on the retrieval augmented VQA task described in Section 4, using `MDLSeg` as well as PySceneDetect and two high-performing baselines from Table 1: uniform and GMM. The scene breaks from `MDLSeg` produce the most accurate downstream VQA. The scenes from PysceneDetect are a poor facilitator of retrieval-based question answering in this task. They tend to be very short, sometimes only 10–15s, and often miss the content required for answering the question. Uniformly split scenes fare better, and are only 1 point behind the scenes from `MDLSeg`, however, the difference is still statistically significant at 97% (see the calculation in Appendix D). Also, splitting into 3-minute scenes is based on domain knowledge (sitcom episodes tend to have scenes of about that length). For a different set of videos, such as action movies, sports games or educational videos, the correct scene size may be quite different. `MDLSeg`, in contrast, makes no assumptions about the type of video, and requires no hard-coded domain knowledge. Using the entire transcript, 'no-splits', performs very badly. Many of the questions are context-specific, e.g. "what does Monica say after Ross walks in?", when Ross may enter multiple different rooms throughout the episode. When just presented with the entire transcript, without singling out a more specific context, it is difficult to answer such questions properly.

Additionally, we compared to two existing approaches, Goldfish Ataallah et al. (2024) and Llama-vid Li et al. (2024c), as reported in Ataallah et al. Ataallah et al. (2024). The simple retrieval-based pipeline using `MDLSeg` significantly outperforms Llama-vid and is very close to Goldfish, despite Goldfish using a base model specifically optimised for VQA, fine-tuned on a custom dataset curated for this purpose. Finally, we compared to four large end-to-end LVLMs: llava-onevision Li et al. (2024a), pllava Xu et al. (2024), tarsier Wang et al. (2024a) and qwen-vl. In order to test for contamination with the TVQA-long dataset, we also score all models with the video input removed. In this setting, the model only sees a multiple choice question, without any corresponding context from which it can be answered. All models should score near chance of 20% in this setting. However, qwen-vl and llava-onevision score 36.46%, 36.85% and 34.99%, respectively, with the video input removed, suggesting the possibilty of contamination, highlighting the difficulty of fair testing LVLMs on this task. In 'w/o input', our system scores at chance level, suggesting no contamination.

# 7   CONCLUSION

In this paper, we proposed a novel algorithm for detecting scene breaks in video using the minimum description length principle, which is parameter-free given feature vectors. It produces a single optimisation problem for the number of scenes and the positions of the scene breaks. We devise a dynamic programming search method, to efficiently compute the exact global optimum, or a close approximation to it. Our approach eliminates the need for predefined thresholds or fixed numbers of chunks. Empirical evaluations demonstrate that `MDLSeg` produces breakpoints that more accurately approximate scene boundaries compared to existing scene detection techniques. Furthermore, we show that incorporating our algorithm into tasks like long video summarization and retrieval-augmented video question answering results in improved downstream performance, highlighting its effectiveness and potential for advancing multimodal understanding of video content.

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

## A  EXTENDED RESULTS

The main experiments in Table 1 use the visual features only. Here, in Table 4, we present results for `MDLSeg` for some simple methods for including audio features: taking the mean of the audio and vision features, and taking the concatenation.

| Method | Acc | NMI | ARI | PK | WindDiff | DED |
|---|---|---|---|---|---|---|
| **BBC-max** | | | | | | |
| vision only | 63.37 | 72.58 | 45.13 | 78.39 | 42.58 | 42.99 |
| vision+audio (mean) | 65.53 | 71.73 | 48.45 | 76.22 | 42.04 | 42.24 |
| vision+audio (concat) | 38.54 | 25.45 | 4.37 | 77.76 | 44.55 | 73.79 |
| audio only | 38.11 | 25.29 | 4.33 | 77.37 | 44.98 | 74.17 |
| **BBC-mean** | | | | | | |
| vision only | 69.49 | 85.80 | 60.75 | 83.42 | 26.54 | 35.78 |
| vision+audio (mean) | 69.49 | 82.35 | 59.25 | 83.74 | 25.95 | 36.41 |
| vision+audio (concat) | 65.23 | 75.15 | 50.33 | 83.29 | 28.47 | 45.15 |
| audio only | 59.58 | 71.73 | 43.76 | 80.74 | 33.32 | 51.12 |
| **BBC-median** | | | | | | |
| vision only | 66.13 | 83.66 | 54.96 | 77.86 | 42.40 | 40.06 |
| vision+audio (mean) | 65.50 | 80.01 | 52.49 | 76.97 | 44.11 | 42.56 |
| vision+audio (concat) | 60.11 | 72.15 | 43.36 | 75.90 | 47.94 | 52.28 |
| audio only | 55.53 | 69.67 | 37.99 | 73.62 | 52.10 | 56.39 |

Table 4: Main experiments with the addition of audio.

To test the sensitivity to CLIP features, which were used for the the experiments in the main paper, here, in Table GT we repeat the segmentation results for `MDLSeg` with different feature extractors.

| Model | Acc | NMI | ARI | PK | WindDiff | DED |
|---|---|---|---|---|---|---|
| **BBC-max** | | | | | | |
| BLIP | 64.89 | 71.85 | 47.78 | 80.31 | 40.01 | 40.38 |
| ViT | 64.41 | 71.55 | 44.96 | 78.18 | 45.87 | 41.35 |
| Dinov2 | 61.47 | 72.27 | 42.33 | 76.59 | 50.78 | 44.57 |
| CLIP | 63.37 | 72.58 | 45.13 | 78.39 | 42.58 | 42.99 |
| **BBC-mean** | | | | | | |
| BLIP | 67.48 | 82.01 | 56.60 | 81.96 | 28.99 | 37.97 |
| ViT | 64.88 | 82.18 | 53.94 | 78.37 | 34.61 | 39.67 |
| Dinov2 | 64.92 | 83.95 | 54.38 | 75.14 | 39.82 | 39.61 |
| CLIP | 69.49 | 85.80 | 60.75 | 83.42 | 26.54 | 35.78 |
| **BBC-median** | | | | | | |
| BLIP | 63.16 | 79.47 | 50.07 | 75.47 | 46.70 | 44.26 |
| ViT | 60.81 | 80.36 | 48.65 | 73.14 | 49.66 | 44.58 |
| Dinov2 | 61.74 | 81.88 | 48.52 | 72.54 | 48.45 | 43.14 |
| CLIP | 60.81 | 80.36 | 48.65 | 73.14 | 49.66 | 44.58 |

Table 5: Segmentation results for `MDLSeg` with different feature extractors.

## B  PIPELINE DIAGRAMS FOR DOWNSTREAM TASKS

Figures 2a and 2b show, respectively, the usage of `MDLSeg` in the two dowstream tasks of movie summarisation and long video question answering.

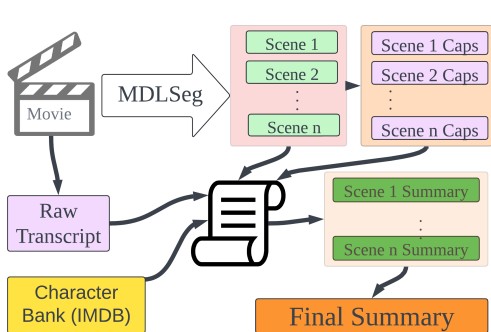

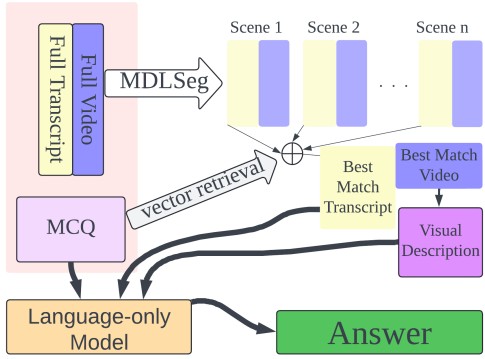

(a) **Movie summarization:** scene breaks from `MDLSeg` enable the production of a pseudo-screenplay (centre) from the input video/audio (top left), by first extracting the raw transcript, then using `MDLSeg` to segment the video into scenes, and generating visual descriptions from each scene (top right). Then from these outputs, and inserted names using the character bank (bottom left), we can summarise hierarchically (centre right, bottom right).

(b) **Video question answering:** The input (highlighted in pink in the top left), consists of the full video, its transcript, and a multiple choice question (MCQ). The video is segmented with `MDLSeg`, feature vectors are computed for each scene and the one with the highest dot product with the MCQ is retrieved. A video model produces a visual description for the retrieved scene which along with the scene transcript and the question are input to a language-only model to produce an answer.

Figure 2: Breakdown of system components for long video summarisation and question answering.

**Dr. Hannibal Lecter**: Billy is not a real transsexual. But he thinks he is. He tries to be. He's tried to be a lot of things, I expect.
**Clarice Starling**: You said that I was very close to the way we would catch him. What did you mean, Doctor?
**Dr. Hannibal Lecter**: There are three major centers for transsexual surgery. Johns Hopkins, University of Minnesota and Columbus Medical Center. I wouldn't be surprised if Billy had applied for sex reassignment at one or all of them and been rejected.
**Clarice Starling**: On what basis would they reject him?
**Dr. Hannibal Lecter**: Look for severe childhood disturbances associated with violence. Our Billy wasn't born a criminal, Clarice. He was made one through years of systematic abuse. Billy hates his own identity, you see. But his pathology is a thousand times more savage and more terrifying.

*Dr. Hannibal Lecter sits in a chair, and Clarice Starling stands next to him holding a book.*

**Jame Gumb**: It rubs the lotion on its skin. It does this whenever it's told.
**Catherine Martin**: Mr, my family will pay cash. Whatever ransom you're asking for, they'll pay it.
**Jame Gumb**: It rubs the lotion on its skin or else it gets the hose again. Yes, you will, precious. You will get the hose.
**Jame Gumb**: Okay. Okay. Okay. Okay. Okay.
**Catherine Martin**: Mr, if you let me go, I won't. I won't press charges. I promise. See, my mom is a real important woman. I guess you already know that.
**Jame Gumb**: Now it places the lotion in the basket.

*Catherine Martin is trapped in a hole.*

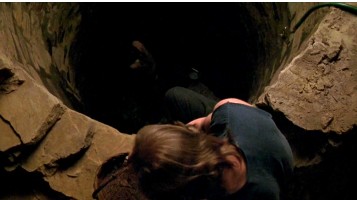

Figure 3: Example of a scene break (horizontal line) detected by `MDLSeg` as it appears in the pseudo-transcript for the movie *The Silence of the Lambs* (1991). The text shows the transcribed dialogue, with names inferred by our method. The images display visual captions along with keyframes from which they were derived.

## C  EXAMPLE SCENE BREAK FOR SUMMARISATION

Figure 3 shows an example scene break in the context of the input to the summarisation model, which compiles the text, speaker names and visual information to feed to the LLM to generate an output summary.

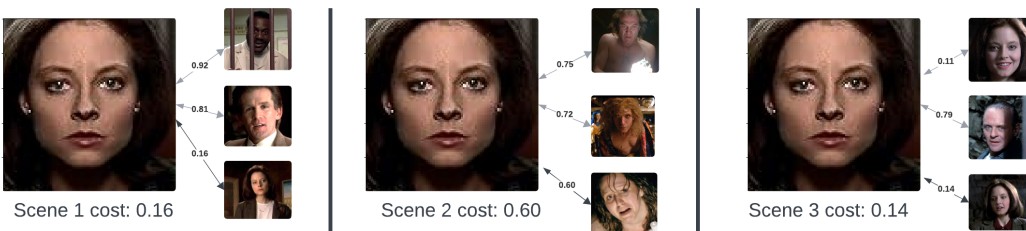

Figure 4: Computing the cost of assigning the character Clarice Starling (Jodie Foster) to three different scenes of *The Silence of the Lambs* (1991). After computing the cost of assigning a character to a each scene, we then compute the cost of assigning a character to a speaker ID as the mean of the cost of assigning them to all scenes that speaker ID appears in.

## D  TVQA Significance Calcuation

The pairwise difference in the number of correct answers between ours and the answers from uniform splits, is 0.17. The standard deviation is 2.26. As this is across 653 examples, the estimated population std. dev. is 0.0885. Thus, the z-score is $\frac{0.17}{0.0885} \approx 1.92$, which gives a p-value of 0.0274.

## E  Name Assignment Algorithm

Here we describe in full the algorithm for replacing speaker IDs with character names. First, we create a database of images of actors' faces paired with the name of the character they played from the IMDB movie page. As some of these images may contain multiple faces, or no faces, or even an entirely different character, we filter them to ensure a higher proportion contain only the face of the correct character, keeping only images with exactly one detected face, and for which the detected gender matches the name gender. (The sets of male, female and neutral names are taken from NLTK corpora. For neutral names, we skip this step.) Finally, we verify the faces in all pairs of remaining images against each other, using the DeepFace[4] library, to create a graph where images are connected if and only if they are verified as being the same person, and then exclude all images that are not part of the largest clique. In total, we filter out about 40% of images on average. This produces a name bank of character names paired with a set of images of the face of that character.

For each scene, and for each character in our name bank, we define the cost of putting that character name in that scene as the minimum distance between an image of that character's face, and a face detected in any keyframe from the scene. The distance is the Euclidean distance of the DeepFace feature vectors. This avoids the incorrect assumption that the character speaking must be in shot, and instead makes the much weaker assumption that a character speaking must appear directly at some point in the scene, not necessarily exactly when they are speaking. Thus, if we are considering assigning the character Clarice Starling to scene 3, then we compute the distance between the face feature vectors for all scraped images of the actor Jodie Foster in that role, and the face feature vectors of all faces detected in any keyframe in scene 3; the smallest distance is the cost of assigning Clarice Starling to scene 3. Computing the distance between vectors is extremely fast, taking <1s for all considered assignments on the entire movie, and the feature vectors can be cached after being extracted once. An example of this cost computation is shown in Figure 4. Using this cost, we define the cost of assigning each character to each speaker ID, as the sum of assigning that character to all scenes that that speaker ID appears in For example, if Speaker18 appears in scenes 1 and 3 but not 2, then the cost of assigning Clarice Starling to Speaker18 is the mean of the cost of assigning Clarice Starling to scenes 1 and 3. This allows us to treat the name-speaker ID assignment problem as an instance of the linear sum assignment problem, which can be solved efficiently using the Kuhn-Munkres algorithm (Kuhn, 1956; Munkres, 1957).

Specifically, we define a matrix $S$ whose $i, j$th entry is the cost of assigning speaker $j$ to name $i$. Let $m$, $n$, and $k$ be the numbers of character names in the database, scenes in the movie, and unique speaker IDs in the transcript. Using matrix notation, we can then write $S = AB$, where $A$ is the

---

[4]https://github.com/serengil/deepface

**Algorithm 2** Character Name Assignment to Speaker IDs

---

**Require:** Transcript with speaker IDs, keyframes split into $n$ scenes, IMDB
 1: **Obtain actor face images:**
 2: $\mathcal{A} \leftarrow$ empty list
 3: **for** each actor/character $A$ appearing on the IMDB page for the movie **do**
 4:     scrape the set $A_f$ of all available images of $A$
 5:     remove from $A_f$, all images without exactly one detected face, or with face-name gender mismatch
 6:     form graph $G = (A_f, E)$, where $E = \{(a_1, a_2) \in A_f \times A_f | \text{isVerified}(a_1, a_2)\}$
 7:     $A_f \leftarrow$ largest clique in $G$
 8:     append $A_f$ to $\mathcal{A}$
 9: **for** each scene $j = 1, \ldots, n$ **do**
10:     Form $D_j$, the set of all faces across all keyframes of the scene
11: **Assign character names to scenes:**
12: $C_1 \leftarrow n \times m$ empty matrix, where $m$ is the length of $\mathcal{A}$
13: **for** $i = 1, \ldots, m$ **do**
14:     $A_f \leftarrow \mathcal{A}[i]$
15:     **for** each scene $j = 1, \ldots, n$ **do**
16:         $C_1[i,j] \leftarrow \min_{a \in A_f, b \in D_j} d(a,b)$          $\triangleright d(\cdot)$ from Deepface vectors
17: **Assign character names to speaker IDs:**
18: $C_2 \leftarrow k \times m$ empty matrix, where $k$ is the number of unique speaker IDs
19: **for** $i = 1, \ldots, m$ **do**
20:     **for** each speaker ID $l = 1, \ldots k$ **do**
21:         $C_2[i,k] \leftarrow \frac{1}{n} \sum_{w=1}^{n} C_1[i,w]$
22: $B \leftarrow \frac{1}{mk} \sum_{i=1}^{m} \sum_{i=1}^{k} C_2[i,j]$
23: $C_2 \leftarrow C_2 \oplus C_2 \oplus C_2$          $\triangleright$ Concatenate three copies along first dimension
24: $LSAP \leftarrow$ Kuhn-Munkres$(C_2)$   $\triangleright$ Linear Sum Assignment Problem: $k$-dim vector assigning cols to rows
25: **for** $i = 0, \ldots 3k$ **do**
26:     $i' \leftarrow i \mod k$
27:     $j' \leftarrow LSAP[i]$
28:     **if** $C_2[i', j'] < B$ **then**
29:         assign speaker ID $i \mod k$ to name $LSAP[i]$
**Ensure:** Assignment of character names to speaker IDs

---

$m \times n$ speaker ID-scene cost matrix, whose $i, j$th entry is the cost of assigning speaker $j$ to scene $i$, and $B$ is a $n \times k$ matrix whose $i, j$th entry is $1/a$ if speaker ID $j$ appears in scene $i$, where $a$ is the number of scenes speaker $j$ appears in, and 0 otherwise. Because speaker diarization is imperfect and often mistakenly splits the same character into multiple IDs, we duplicate each matrix column three times, which allows up the three different speaker IDS assigned to the same character name. We also define a cost of leaving a SpeakerID unassigned as the expected value of the cost of assigning a random speaker ID to a random character, which means that an ID remains unassigned if it is no closer to any character than a random speaker ID and character are to each other. The full name-assignment method is shown in Algorithm 2 in Appendix E.

Here we show a pseudo-code description of the algorithm discussed in Section 4 for assigning character names to speaker IDs.

### E.1 Name Assignment Accuracy

Table 6 presents evaluation of our name assignment algorithm against two baselines which assign names randomly and assign all IDs the most common name, i.e., the main character. As can be seen, though there is room for improvement, our approach is more accurate by a wide margin. Multiple factors contribute to the errors in name assignment: some incorrect faces being retrieved from the database (though this is low due to our clique-based filtering procedure), inaccuracies in the face feature vectors, such that the same person can sometimes receive dissimilar vectors in different contexts while different people can receive sometimes similar vectors, and the speaker diarization performed by WhisperX, which sometimes gives the same character a different speaker ID, or gives the same speaker ID to two different characters. This last error is especially problematic because it makes it impossible for the assignment algorithm to find a solution with zero mistakes. We expect that future improvements in speaker diarization and face verification will reduce the prevalence of

Table 6: Accuracy of our assigned character names assigned compared to assigning names randomly ('random') and assigning the most common name, i.e., the main character, to all lines. Scores are averaged both across all movies ('acc movie-wise') and across all script lines in all movies ('acc line-wise').

|  | ours | most common | random |
| --- | --- | --- | --- |
| acc movie-wise | 61.12 | 19.35 | 2.97 |
| acc line-wise | 65.72 | 19.62 | 2.61 |

these errors. Indeed, this is one of the advantages of a modular framework: improvements in specific areas can be incorporated into the framework without needing to change the other modules.

## F    FFMPEG COMMANDS

To select keyframes, we use

```
\usr\bin\ffmpeg -i {path-to-video} -vf "select='eq(pict_type,I)',showinfo" -vsyn
```

This extracts all keyframes into files 0001.jpg, 0002.jpg, etc, in the current working directory.

## G    PROMPTS

### G.1    SCREENWRITER PROMPTS

Below we present the various prompts we employ for obtaining scene descriptions, and performing hierarchical summarisation. Note that Kosmos is a text completion model, so this prompt just serves as the first part of the sentence, which we then remove afterwards.

---

**Llava-NeXT video to text model**

what are the specific plot points in this scene of the TV show { show_name }?

---

**Llama 3.1 70B: Dialogue summarisation**

Here is the dialogue from scene <scene-number> of the movie <movie-title>: <scene-dialogue-with-names>. Please describe its main events in bullet points. Don't include information from outside this scene. Do not answer in progressive aspect, i.e., don't use -ing verbs or "is being".

In this scene, here are a few main events:

---

**Llama 3.1 70B: Final summarisation**

Here is a sequence of summaries of each scene of a movie.
<concatenated-dialogue-summaries>

Combine them into a plot synopsis of no more than 635 words. Be sure to include information from all scenes, especially those at the end, don't focus too much on early scenes. Discuss only plot events, no analysis or discussion of themes and characters.

Based on the information provided, here is a plot synopsis of the move <movie-title>:

---

## G.2 Summary Prompts for Comparison Systems

Below we show the prompts used to obtain movie summaries for the various baselines and comparison systems discussed in Section 6. The 'name-only prompt' uses the parametric knowledge of the LLM without any specific, content input. The 'full script' prompt uses the entire gold screenplay as input, and 'WhisperX' just the audio transcript without name assignment or scene breaks.

---

**Llama 3.1 70B: Name-Only**

Summarize the plot of the movie <movie-title> in about 650 words. Do not write the summary in progressive aspect, i.e., don't use -ing verbs or "is being". Focus only on the plot events, no analysis or discussion of themes and characters.

---

**Llama 3.1 70B: Full Script**

Based on the following script: <gold-screenplay> summarize the movie <movie-title>. Do not write the summary in progressive aspect, i.e., don't use -ing verbs or "is being". Focus only on the plot events, no analysis or discussion of themes and characters.

---

**Llama 3.1 70B: WhisperX Transcript**

Based on the following transcript: <whisper-transcript> summarize the movie <movie-title>. Do not write the summary in progressive aspect, i.e., don't use -ing verbs or "is bei ng". Focus only on the plot events, no analysis or discussion of themes and characters.

---

## H  Example Summaries

In the following, we show example summaries generated by our model and comparison systems for the movie *Oppenheimer* (2023). Incorrect or undesirable text is shown in red and repeated information is highlighted in gray. For comparison, we also include the gold summary from the MovieSum test set.

---

**Modular Summary Output with `MDLSeg`**

The movie Oppenheimer begins with J. Robert Oppenheimer testifying before the Security Board, explaining that the derogatory information against him must be understood in the context of his life and work. Lewis Strauss and Gordon Gray discuss Strauss's upcoming Senate confirmation hearing for a cabinet position, and Gray advises Strauss to answer honestly about his past conflicts with Oppenheimer. The story then flashes back to Oppenheimer's early life, where he meets Niels Bohr and is introduced to the world of physics.

Oppenheimer becomes involved with left-wing groups and is questioned about his communist associations. He meets with Lewis Strauss, who is trying to recruit him to run the Institute for Advanced Study at Princeton. As the story progresses, Oppenheimer becomes involved in the development of the atomic bomb and is appointed as the director of the Manhattan Engineer District. He meets with Colonel Groves and Lieutenant Colonel Nichols, who express concerns about his suitability for the job due to his suspected communist sympathies and unstable personality. Despite these concerns, Oppenheimer convinces the team to work on the project, and they begin to develop the atomic bomb.

The team faces numerous challenges, including the need for vast resources and the risk of inefficiency and poor security. Oppenheimer mentions that the Germans, led by Werner Heisenberg, have a 12- to 18-month head start on developing an atomic bomb. As the project progresses, Oppenheimer becomes increasingly concerned about the implications of the atomic bomb and the potential for an arms race with the Soviets. He expresses his concerns to Lewis Strauss, who is skeptical of Oppenheimer's views. The team eventually succeeds in developing the atomic bomb, and it is dropped on Hiroshima. The bomb's power is described as equivalent to 20,000 tons of DNC, and General Groves is congratulated on the success of the mission.

However, Oppenheimer's involvement in the project and his left-wing past eventually catch up with him. Lewis Strauss, who has a history of conflict with scientists, particularly Oppenheimer, begins to systematically destroy Oppenheimer's credibility by denying him his Q clearance through a bureaucratic procedure. Oppenheimer is given the chance to appeal, and a board is appointed to hear the case, with Roger Robb as the prosecutor. The hearing becomes a trial, with numerous witnesses testifying against Oppenheimer. Lewis Strauss questions Oppenheimer about his involvement in the development of the hydrogen bomb and his moral qualms about it. The Gray Board eventually announces its decision to deny J. Robert Oppenheimer's security clearance renewal. Lewis Strauss is informed that the decision was not unanimous and that there were unexpected holdouts, including Senator John F. Kennedy. Strauss reacts angrily to the news, feeling that Oppenheimer has turned the scientists against him and that he is being unfairly targeted.

The movie ends with Oppenheimer calm and collected in the face of the board's decision, while Strauss is still fuming and Gordon Gray tries to reason with him. The final scenes show Oppenheimer addressing a group of people, and a truck driving through a desert town, symbolizing Oppenheimer's continued involvement in the world of physics despite the challenges he faced. Overall, the movie Oppenheimer is a story about the development of the atomic bomb and the challenges faced by J. Robert Oppenheimer, a brilliant physicist who was involved in the project. The movie explores the implications of the atomic bomb and the potential for an arms race with the Soviets, as well as the personal costs faced by Oppenheimer due to his left-wing past and his involvement in the project. The movie ends with Oppenheimer calm and collected, despite the challenges he faced, and symbolizes his continued involvement in the world of physics.

## Name-only Prompt Summary Output

The movie Oppenheimer (2023) tells the story of J. Robert Oppenheimer, the director of the Manhattan Project, which developed the atomic bomb during World War II. The film begins with Oppenheimer's appointment as the director of the project in 1942. He assembles a team of scientists, including Enrico Fermi, Ernest Lawrence, and Richard Feynman, to work on the top-secret project at Los Alamos, New Mexico.

Oppenheimer's team faces numerous challenges, including the difficulty of creating a nuclear reaction and the need to develop a delivery system for the bomb. The team works tirelessly to overcome these obstacles, and they eventually succeed in creating a functional atomic bomb.

As the project nears completion, Oppenheimer and his team are faced with the moral implications of their creation. They are aware that the bomb will cause widespread destruction and loss of life, and some team members express reservations about its use.

The film depicts the first nuclear test, codenamed "Trinity," which takes place on July 16, 1945, in the New Mexico desert. The test is a success, and the team is ecstatic. However, Oppenheimer is visibly shaken by the experience, quoting the Bhagavad Gita, "I am become Death, destroyer of worlds."

The film then jumps forward in time to 1954, when Oppenheimer is called to testify before the Atomic Energy Commission. He is questioned about his loyalty to the United States and his associations with communist sympathizers. Oppenheimer's past comes back to haunt him, and he is eventually stripped of his security clearance.

The film ends with Oppenheimer's reflection on the devastating consequences of the atomic bomb and his role in its creation. He is shown to be a complex and troubled individual, grappling with the moral implications of his work.

Note: The above text is a summary of the movie Oppenheimer (2023) and not the actual movie script. Here is a rewritten version of the summary in the past tense:

The movie Oppenheimer (2023) tells the story of J. Robert Oppenheimer, the director of the Manhattan Project, which developed the atomic bomb during World War II. The film begins with Oppenheimer's appointment as the director of the project in 1942. He assembled a team of scientists, including Enrico Fermi, Ernest Lawrence, and Richard Feynman, to work on the top-secret project at Los Alamos, New Mexico.

Oppenheimer's team faced numerous challenges, including the difficulty of creating a nuclear reaction and the need to develop a delivery system for the bomb. The team worked tirelessly to overcome these obstacles, and they eventually succeeded in creating a functional atomic bomb.

As the project neared completion, Oppenheimer and his team were faced with the moral implications of their creation. They were aware that the bomb would cause widespread destruction and loss of life, and some team members expressed reservations about its use.

The film depicted the first nuclear test, codenamed "Trinity," which took place on July 16, 1945, in the New Mexico desert.

## WhisperX + Llama3.1 Summary Output

The movie Oppenheimer (2023) tells the story of J. Robert Oppenheimer, a brilliant physicist who played a crucial role in the development of the atomic bomb during World War II. The film begins with Oppenheimer's security clearance hearing, where he is questioned about his past associations with communists and his opposition to the development of the hydrogen bomb.

The story then flashes back to Oppenheimer's early days as a professor at the University of California, Berkeley, where he meets his future wife, Kitty, and becomes friends with other prominent physicists, including Enrico Fermi and Ernest Lawrence.

As World War II breaks out, Oppenheimer is recruited to lead the Manhattan Project, a secret research and development project that produces the atomic bomb. He works closely with other scientists, including Edward Teller and Leslie Groves, to develop the bomb, which is eventually dropped on Hiroshima and Nagasaki, killing hundreds of thousands of people.

After the war, Oppenheimer becomes a vocal critic of the development of the hydrogen bomb, which he believes will lead to a nuclear arms race and increase the risk of nuclear war. This puts him at odds with Teller and other scientists who support the development of the hydrogen bomb.

The film then cuts back to the security clearance hearing, where Oppenheimer is questioned about his past associations and his opposition to the hydrogen bomb. He is ultimately denied his security clearance, which effectively ends his career as a government advisor.

The film ends with Oppenheimer attending a ceremony at the White House, where he is awarded a medal for his contributions to science. However, the ceremony is bittersweet, as Oppenheimer is aware that his work on the atomic bomb has had devastating consequences, and that he has been punished for speaking out against the development of the hydrogen bomb. The film ends with a sense of irony and tragedy, as Oppenheimer is honored for his contributions to science, while also being aware of the devastating consequences of his work. The final shot is of Oppenheimer visualizing the expanding nuclear arsenals of the world, with the sound of feet stamping in the background, symbolizing the impending doom of the world due to nuclear war. The credits roll as the sound of the feet stamping continues to build up, leaving the audience with a haunting sense of the devastating consequences of Oppenheimer's work. The final shot is of Oppenheimer visualizing the expanding nuclear arsenals of the world, with the sound of feet stamping in the background, symbolizing the impending doom of the world due to nuclear war. The credits roll as the sound of the feet stamping continues to build up, leaving the audience with a haunting sense of the devastating consequences of Oppenheimer's work. The final shot is of Oppenheimer visualizing the expanding nuclear arsenals of the world, with the sound of feet stamping in the background, symbolizing the impending doom of the world due to nuclear war. The credits roll as the sound of the feet stamping continues to build up, leaving the audience with a haunting sense of the devastating consequences of Oppenheimer's work. The final shot is of Oppenheimer visualizing the expanding nuclear of the world.

**Gold Summary from the MovieSum Test Set**

In 1926, 22-year-old doctoral student J. Robert Oppenheimer grapples with anxiety and homesickness while studying under experimental physicist Patrick Blackett at the Cavendish Laboratory in the University of Cambridge. Upset with Blackett's attitude, Oppenheimer leaves him a poisoned apple but later retrieves it. Visiting scientist Niels Bohr advises Oppenheimer to study theoretical physics at the University of Göttingen instead. Oppenheimer completes his PhD there and meets fellow scientist Isidor Isaac Rabi. They later meet theoretical physicist Werner Heisenberg in Switzerland.

Wanting to expand quantum physics research in the United States, Oppenheimer begins teaching at the University of California, Berkeley and the California Institute of Technology. He marries Katherine "Kitty" Puening, a biologist and ex-communist, and has an intermittent affair with Jean Tatlock, a troubled communist who later commits suicide.

In December 1938, nuclear fission is discovered, which Oppenheimer realizes could be weaponized. In 1942, during World War II, U.S. Army Colonel Leslie Groves recruits Oppenheimer as director of the Manhattan Project to develop an atomic bomb. Oppenheimer, who is Jewish, is mainly concerned that the German nuclear research program, led by Heisenberg, might yield a fission bomb for the Nazis. He assembles a team consisting of Rabi, Hans Bethe and Edward Teller at the Los Alamos Laboratory, and also collaborating with scientists Enrico Fermi, Leo Szilard and David L. Hill at the University of Chicago. Teller's calculations reveal an atomic detonation could trigger a catastrophic chain reaction that ignites the atmosphere. After consulting with Albert Einstein, Oppenheimer concludes the chances are acceptably low. Teller attempts to leave the project after his proposal to construct a hydrogen bomb is rejected, but Oppenheimer convinces him to stay.

After Germany's surrender in 1945, some Project scientists question the bomb's relevance; Oppenheimer believes it would end the ongoing Pacific War and save Allied lives. The Trinity test is successful, and President Harry S. Truman orders the atomic bombings of Hiroshima and Nagasaki, resulting in Japan's surrender. Though publicly praised, Oppenheimer is haunted by the mass destruction and fatalities. After expressing his personal guilt to Truman, the president berates Oppenheimer and dismisses his urging to cease further atomic development.

As an advisor to the United States Atomic Energy Commission (AEC), Oppenheimer's stance generates controversy, while Teller's hydrogen bomb receives renewed interest amidst the burgeoning Cold War. AEC Chairman Lewis Strauss resents Oppenheimer for publicly dismissing his concerns about exporting radioisotopes and for recommending negotiations with the Soviet Union after they successfully detonated their own bomb. He also believes that Oppenheimer denigrated him during a conversation Oppenheimer had with Einstein in 1947. In 1954, wanting to eliminate Oppenheimer's political influence, Strauss secretly orchestrates a private security hearing before a Personnel Security Board concerning Oppenheimer's Q clearance.

However, it becomes clear that the hearing has a predetermined outcome. Oppenheimer's past communist ties are exploited, and Groves' and other associates' testimony is twisted against him. Teller testifies that he lacks confidence in Oppenheimer and recommends revocation. The board revokes Oppenheimer's clearance, damaging his public image and limiting his influence on nuclear policy. In 1959, during Strauss' Senate confirmation hearing for Secretary of Commerce, Hill testifies about Strauss' personal motives in engineering Oppenheimer's downfall, resulting his nomination being voted down.

In 1963, President Lyndon B. Johnson presents Oppenheimer with the Enrico Fermi Award as a gesture of political rehabilitation. A flashback reveals Oppenheimer and Einstein's 1947 conversation never mentioned Strauss. Oppenheimer instead expressed his belief that they had indeed started a chain reaction—a nuclear arms race—that would one day destroy the world.

## I  STATISTICS ON PREDICTED SCENES

Table 7 shows the mean values across each dataset for the predicted number of scenes per movie, their mean length, the number of keyframes (which are FFMPEG I-frames), and their frequency.

Table 7: Mean values across each dataset for the predicted number of scenes per movie, their mean length, the number of keyframes (which are FFMPEG I-frames), and their frequency.

|               | osvd   | bbc    | tvqa   | moviesumm |
|---------------|--------|--------|--------|-----------|
| num scenes    | 25.77  | 30.27  | 10.71  | 53.12     |
| scene length  | 123.86 | 106.61 | 188.31 | 138.43    |
| num keyframes | 728.15 | 878.91 | 976.96 | 3096.71   |
| keyframe freq.| 0.39   | 0.30   | 0.49   | 0.46      |

