# OpenReview forum: "A Parameter-free Scene Detection Algorithm for Vision and Language Understanding"
_ICLR.cc/2026/Conference — ICLR 2026 Conference Withdrawn Submission_

### Official Review · Reviewer_2KJT · 2025-10-22

**Soundness:** 2
**Presentation:** 1
**Contribution:** 2
**Rating:** 2
**Confidence:** 4

**Summary:**

This paper proposes a novel method for video scene segmentation based on the Minimum Description Length (MDL) principle. By leveraging the MDL property, the proposed approach achieves an elegant balance between two conflicting objectives in scene segmentation: ensuring intra-scene frame similarity while controlling the total number of scenes. Notably, this balance is attained without any manual parameter tuning or training. Experiments demonstrate that our method yields more accurate scene boundaries, particularly in long video settings, and further enhances the performance of two downstream tasks—long video summarization and long video question answering (long video QA).

**Strengths:**

1.	The paper proposes a novel MDL-based method for video scene segmentation that is both parameter-free and training-free. This design elegantly balances two conflicting objectives in clustering-based problems—preserving intra-scene frame similarity while controlling the overall number of scenes. The simplicity and conceptual clarity of the approach are commendable.

2.	The exploration of downstream tasks, including long video summarization and question answering, demonstrates improved performance facilitated by the proposed scene segmentation method. This indicates that better scene segmentation can effectively enhance long-video understanding.

**Weaknesses:**

Despite the strengths demonstrated above, this paper should be rejected for the following main reasons:

1.	The motivation and the specific problems addressed in this study are not clearly articulated, which raises questions about the significance and necessity of the proposed method.

2.	The effectiveness of independent chunk understanding in video summarization, as presented in lines 35–37, is questionable. Effective video summarization requires a holistic comprehension of the overall narrative and structural flow. Processing individual segments in isolation may neglect essential global dependencies and disrupt contextual coherence.

3.	The method is presented in a confusing manner without sufficient preliminaries or clear equations to define the cost of a partition. The unclear relationships among frames, scenes, partitions, and videos further obscure the overall design.

4.	There is a concern regarding the fine-grained segmentation and generalization capabilities of the method while dealing with scenes that share similar statistical characteristics but differ in semantic content, which often occurs in transitional shots. The MDL-based approach may fail to correctly segment such scenes due to the MDL principle’s emphasis on minimizing description length.

5.	The demonstrations of the two downstream tasks in Section 4 are unclear. The claim in line 243 that the system “requires only video input” neglects the character bank from IMDB. In line 250, the phrase “timestamps in the transcript” is ambiguous, raising the question of whether these timestamps are aligned with the boundaries of the segmented scenes. Moreover, it is unclear in line 254 how cosine similarity is computed between a multi-frame scene and a single question.

6.	The experimental section is insufficient in several aspects:

(1) Lines 194–196 introduce a hyperparameter L, which is stated to achieve the best performance when set to approximately 10 minutes; however, no explicit ablation study is provided to support this claim.

(2) Commonly used evaluation metrics for video scene segmentation—such as AP, mIoU, AUC-ROC, and F1 [1, 2, 3]—should also be considered and compared in the experiments.

(3) The conclusion presented in lines 456–459 lacks experimental validation. It would be beneficial to include comparative experiments on different sets of videos to substantiate this conclusion.

7.	The analysis of the “w/o names” setting in paragraph Summarization of Section 6 is confusing, as this setting is discussed in the text but not shown in Table 2.

8.	The purpose of the “w/o input” setting in Table 3 is debatable, as the proposed method does not involve any training process. Therefore, it seems unnecessary to discuss the potential contamination issue within the proposed system.

Besides, the paper is imprecise and unpolished:

1.	The title “Video Understanding” in the second paragraph of the Related Work section is broader than the actual content of the paragraph. As outlined in [4], video understanding generally includes three major categories: video content understanding, descriptive understanding, and video content generation and manipulation. However, the paragraph primarily discusses video captioning, video summarization, and video question answering, which belong specifically to descriptive understanding tasks.

2.	The target task is ambiguously defined, being inconsistently referred to as video segmentation, scene segmentation, and scene detection.

3.	The coverage of prior work is incomplete. Despite the claim in Section 6 that all existing scene segmentation methods are included in Section 2, relevant studies such as [5, 6] are omitted.

4.	Lines 370–377 focus on video summarization rather than scene detection and should be relocated to the subsequent paragraph for clarity.

Presentation and Formatting Issues:

1.	There are extensive citation formatting errors that need to be corrected. For example, author-year references are sometimes written without parentheses.

2.	The frame size in footnote on Page 2 can be better written as “1024\times 1024”.

3.	“scaped” in line 211 should be “scraped”.

4.	There should be a space after the full stop symbol in line 101 and line 316.

5.	The table in lines 378–394 lacks a caption and the whole table appears to have been deleted.

6.	“eTable 2” in line 426 should be “Table 2”.

7.	Citation “Ataallah et al.” in line 464 is repeated.

8.	Method qwen-vl in line 468 should be cited.

[1] Mun, Jonghwan, et al. "Bassl: Boundary-aware self-supervised learning for video scene segmentation." Proceedings of the Asian Conference on Computer Vision. 2022.

[2] Wu, Haoqian, et al. "Scene consistency representation learning for video scene segmentation." Proceedings of the IEEE/CVF conference on computer vision and pattern recognition. 2022.

[3] Tan, Jiawei, et al. "Neighbor Relations Matter in Video Scene Detection." Proceedings of the IEEE/CVF Conference on Computer Vision and Pattern Recognition. 2024.

[4] Madan, Neelu, et al. "Foundation models for video understanding: A survey." arXiv preprint arXiv:2405.03770 (2024).

[5] Chen, Shixing, et al. "Shot contrastive self-supervised learning for scene boundary detection." Proceedings of the IEEE/CVF conference on computer vision and pattern recognition. 2021.

[6] Islam, Md Mohaiminul, et al. "Efficient movie scene detection using state-space transformers." Proceedings of the IEEE/CVF conference on computer vision and pattern recognition. 2023.

**Questions:**

Considering the weaknesses mentioned above, it would be appreciated if the authors could address the following questions:

1.	Please clarify the motivation of the study and specify the particular problems it aims to address.

2.	When processing individual segments in isolation, does the method neglect essential global dependencies and potentially disrupt contextual coherence?

3.	Please clarify the relationships among frames, scenes, partitions, and videos within the proposed framework.

4.	Please provide an explicit definition or formulation of the final cost for a given partition.

5.	Can the proposed method accurately segment scenes that share similar statistical characteristics but differ in semantic content (e.g., transitional shots)?

6.	Please clarify the demonstrations of the two downstream tasks discussed in Section 4, as mentioned in the fifth weakness item.

7.	Lines 367–369 indicate that the proposed method achieves a similar inference speed to more complex methods such as BASSL and NeighborNet. However, Table 1 shows that MDLSeg performs similarly to these methods on the OVSD dataset, but significantly outperforms them on the BBC dataset (approximately 24× faster than BASSL and 2.5× faster than NeighborNet). Please explain the reason for this discrepancy.

8.	Please clarify the “w/o names” setting mentioned in Table 2.

9.	Please clarify the purpose of the “w/o input” setting in Table 3, given that the proposed method does not involve a training process.

---

### Official Review · Reviewer_XYGf · 2025-10-30

**Soundness:** 3
**Presentation:** 2
**Contribution:** 3
**Rating:** 4
**Confidence:** 4

**Summary:**

This paper introduces MDLSeg, a novel parameter-free video segmentation algorithm grounded in the Minimum Description Length (MDL) principle. The method segments videos into contiguous scenes by minimizing the total bit cost of representing frame features, balancing intra-segment compactness and the number of segments. MDLSeg requires no user-defined thresholds, number of scenes, or training data, making it data- and parameter-free once frame features are provided. Dynamic programming is used to solve the formulation.

**Strengths:**

+ Applying the MDL principle to contiguous video segmentation is conceptually elegant and mathematically principled. It provides a unified criterion for determining both scene boundaries and their number without manual hyperparameters.
+ Parameter-free and general, MDLSeg requires no thresholds or tuning, and no labeled training data.
+ The dynamic programming approach provides an efficient and globally optimal (or near-optimal) solution under contiguity constraints.
+ Good performance is achieved.

**Weaknesses:**

1) This method determines both scene boundaries and their number without manual hyperparameters. How is this achieved? MDL (-log_2 p(v)) measures the bits for each segment. Whenever a partitioned segment is overlong, will the MDL be larger? In other word, when a partitioned segment is over short, would the MDL be smaller? Why the optimization partition leads to the smallest MDL? Is there theoretical support?


2) The define of cost is not clear. Is 2dm - log_2 p(v) the cost? Does 2dm play the role of punishing over segment? How to trade off 2dm and -log_2 p(v)? For different feature encoding manner, d would different a lot.


3) The contribution is incremental.  Using DP to address optimization problem is widely used in different field. If the cost is the contribution, there is a lack of deep analysis and explanation of it.

**Questions:**

See weakness part.

---

### Official Review · Reviewer_FjJv · 2025-11-05

**Soundness:** 2
**Presentation:** 3
**Contribution:** 2
**Rating:** 4
**Confidence:** 4

**Summary:**

The authors propose MDLSeg, a novel algorithm for segmenting videos into contiguous chunks based on the Minimum Description Length (MDL) principle, coupled with a dynamic programming search. Given the input feature vectors, the proposed method is entirely parameter-free, thus removing the need to specify thresholds, scene numbers and scene sizes.
The authors compare the proposed scene boundary detection method with existing methods including PySceneDetect and deep learning models. Further the performance of the scene boundary detection method is evaluated on two downstream tasks including hierarchical long video summarization  and retrieval-augmented video Question Answering (QA).

**Strengths:**

* Parameter free design of proposed method MDLSeg that does not require setting any threshold or specifying number of scenes and training on video-boundary based datasets.
* Superior performance of MDLSeg when compared to deep-learning and algorithm-based methods on video scene segmentation.
* Effective usage of MDLSeg for scene segmentation followed by downstream usage in video summarization, retrieval-augmented video question answering.

**Weaknesses:**

* The runtime and quality of the proposed MDLSeg method is dominated by the visual feature extraction stage.
* In **appendix table 5**, the authors show the variation in performance based on the various vision feature extractors. While the VL models included for visual feature extraction are dual-stream (CLIP, BLIP), have the authors considered features from single-stream VL models (with LLM alignment) like **Qwen**, **LLava**, **InternVL** ?
* Inclusion of multimodal features (vision +audio) results in the degradation of the performance (**Appendix Table 4**). Better multimodal fusion operations are needed based on trainable networks.
* Why were closed (gemini) and open models (**Video-XL** (https://openaccess.thecvf.com/content/CVPR2025/papers/Shu_Video-XL_Extra-Long_Vision_Language_Model_for_Hour-Scale_Video_Understanding_CVPR_2025_paper.pdf), **Long-ViLA** (https://arxiv.org/abs/2408.10188))  not considered for summarization tasks as baselines ?
* In terms of downstream utility, ego-centric applications through datasets like **Ego4D-HCap**, **Ego-Schema** and video captioning tasks on **MSRVTT**, **YouCook2** can be considered.

**Questions:**

* In the long-video summarization task, why was face tracking not considered for assigning character IDs ?

---

### Official Review · Reviewer_xFc4 · 2025-11-11

**Soundness:** 2
**Presentation:** 3
**Contribution:** 1
**Rating:** 2
**Confidence:** 5

**Summary:**

This paper proposes to adopt the minimum-description-length-based (MDL-based) clustering algorithm to detect the video scene breaks, where the clustering is applied upon the keyframes (obtained via ffmpeg I-frames), with noting that such clustering algorithm is parameter-free (as the number of clusters can be automatically determined) and training-free (while the keyframe features are extracted by pretrained visual encoders, e.g. CLIP). The video segments resulted from the detected scene breaks are further used as the basis to perform two downstream tasks, including video summarization and retrieval-augmented video question answering. The proposed method is experimentally shown to provide superior performance upon scene break detections, as well as the two downstream tasks, compared with the corresponding baselines in different tasks.

**Strengths:**

+ The paper is easy to follow and well organized.
+ The proposed method not only achieves the superior performance in scene break detection and also contributes to the improvement in the downstream tasks.

**Weaknesses:**

- The main contribution of this submission is to adopt the minimum-description-length-based (MDL-based) clustering algorithm to detect the video scene breaks, in which the detected scenes (i.e. the video partitions/segments based on the detected scene breaks) are further used to perform the downstream tasks (i.e. video summarization and retrieval-augmented video question answering). First of all, the main contribution (i.e. MDL-based scene break detection) in this submission has been explored in the prior work from Mahon et al., A modular approach for multimodal summarization of TV shows, ACL 2024, hence there is no scientific novelty of this submission; Second, the initialization of keyframes plays an important role for the proposed scene break detection, however, there lacks for the corresponding discussion nor extensive experiments to perform the investigation. Moreover, it would be interesting to see if some of the baselines in Table 1 could be leveraged for providing the keyframe initialization; Third, the downstream tasks are more or less the direct applications, with some improvements being additionally achieved by the engineering techniques (e.g. replacing speaker IDs with character names), hence they can not be treated as the scientific contributions. Moreover, there misses a baseline of adopting the oracle scene breaks, for demonstrating that the optimal/groundtruth scene breaks are able to contribute to the best performance in downstream tasks.

**Questions:**

The authors should address the aforementioned weaknesses, including the lack of scientific contribution/novelty, the missing discussion upon the initialization of keyframes, and the baselines of adopting oracle scene breaks to perform the downstream tasks, in the rebuttal.

---

### Note · Authors · 2025-11-20

I have read and agree with the venue's withdrawal policy on behalf of myself and my co-authors.